# Towards the Effectiveness of 3D Printing on Tactile Content Creation for Visually Impaired Users

**DOI:** 10.3390/polym15092180

**Published:** 2023-05-03

**Authors:** Gutenberg Barros, Walter Correia, João Marcelo Teixeira

**Affiliations:** 1Ciência e Tecnologia de Pernambuco (IFPE-Campus Recife), Instituto Federal de Educação, Recife 50740-545, Brazil; gutenbergbarros@recife.ifpe.edu.br; 2Departamento de Design, Universidade Federal de Pernambuco, Recife 50670-901, Brazil; wfmc@cin.ufpe.br; 3Departamento de Eletrônica e Sistemas, Universidade Federal de Pernambuco, Recife 50670-901, Brazil

**Keywords:** 3D printing, additive manufacturing, visually impaired people, accessibility

## Abstract

We have conducted research on how tactile content is created for visually impaired individuals. From the data collected, an experiment was developed and applied. It investigated alternative materials to serve as a basis for the use of 3D printing to reduce production costs. It also evaluated the adherence of different values of width, height, and angles of the contour lines, as well as different geometric shapes and top/bottom fill patterns on these materials. The results show it is possible to use cellulose-based materials weighing between 120 g/m^2^ and 180 g/m^2^ to support the prints instead of making a base for the information, with gains up to 40 times in production time and up to 29 times in the consumption of materials if there is no need to fold the manufactured content. Based on visually impaired every-day activities such as locating and following a line (exploration), discerning different textures (tactile discrimination), identifying figures (picture comprehension), and locating copies of them (spatial comprehension), the ideal line widths for 3D printing adherence regarding tactile content creation were found to be between 0.8 mm and 1.2 mm, while 0.4 mm was the maximum height that did not compromise adherence. When bending the 3D printed material on the surface, we found that lines with angles between 0° and 20° from the bending direction could keep their adherence as well. The shapes must receive a small rounding at the corners and preferably align themselves with the mentioned angles. The top/bottom fill patterns did not affect adhesion. The infill can be used as a texture generator and should be adjusted to densities of 10% to 50%, or 10% to 90% when combined with other textures. In the first case, users were able to perceive differences in the tactile content whenever a single infill pattern was used. In the latter, combining two infill patterns leads to a more discriminating surface, resulting in a higher number of textures to be used in tactile content production (analogous to the number of colors used in an image for a person with no visual impairment).

## 1. Introduction

Additive manufacturing, as a manufacturing technology, is a process by which parts are created directly from three-dimensional mathematical models generated on computers and sent directly to the manufacturing machine, which materializes the objects through controlled and overlapping addition (not subtraction) of materials [1,2].

This property of adding materials layer by layer allows for the manufacture of interior elements as well as fittings in a single step. However, due to technological and even conceptual limitations, 3D printing is generally not directed towards mass production [1,3,4,5].

With its intrinsic advantages and disadvantages, 3D printing has gained great interest from society due to the proliferation of low-cost devices [6] (most of them using FFF—Fused Filament Fabrication—technology), as well as the use of open-source software. Oropallo and Piegl [7] remind us that media coverage of the subject tends to be sensationalistic and, based on popular articles, creates the impression that this technology can be the solution to all problems. Exotic applications, such as bio-printing (manufacturing by precise layer positioning of biological, chemical, and living cell materials aimed at constructing three-dimensional biological structures [8]), appear in the news every semester. Other unconventional but important 3D printing applications may include Bio-Printing operations [9] and bone tissue engineering [10].

A noticeable advantage of 3D printing is that its source files can be customized to create unique and personalized designs while still maintaining the precision and consistency of industrial-quality production. Furthermore, the models can be easily shared digitally and reproduced anywhere in the world and even beyond earth [11].

Associating additive manufacturing with populations with specific needs, including those with some type of physical disability, can generate new possibilities of use, leading to personalized products with potentially encouraging results that can be easily replicated anywhere. Many initiatives already exist in this sense, such as the “e-NABLE” foundation community (http://www.enablecommunityfoundation.org, accessed on 30 April 2023), whose website serves as a central point for the voluntary submission and free collection of three-dimensional model files for the 3D printing of low-cost prostheses for arms and hands. “exiii” is another example: a Japanese incubated company, whose three owners created three adaptable models of hand and forearm prostheses with electronic activation and 50 times lower cost than the more traditional models [12].

On the other hand, images, such an important component of science, technology, engineering, mathematics (STEM), and education, do not receive the same treatment. Braille, a writing method created by Louis Braille in 1824 that has received several revisions and additions over the years, is now the primary non-auditory means of communication for users with reduced or absent visual capacity. It uses a 2 by 3 point matrix, called a cell, to represent letters, numbers, punctuation, and musical signs through the existence and relative position of these points. It can describe a scene, but words hardly reach the same potential as a drawing to convey an idea.

Restricting artistic topics, Braille cannot convey the shapes of the image. Even audio description uses words to try to translate the meaning of the figure, which is basically the same approach, which proves ineffective in presenting the nuances and “errors” of a sketch, a powerful tool to stimulate new ideas.

Thus, considering the idea of reducing the costs of 3D printers, their capacity to materialize complex and customized shapes, and the fact that they are more versatile than a Braille printer, three-dimensional rapid prototyping has the potential to overcome the problem of visual idea transmission [13].

In this direction, Marc Dillon has recreated works of art such as the Mona Lisa and Van Gogh’s sunflowers in three-dimensional versions using 3D printing with sand and binders [14]. Also, the Spanish startup Durero developed a process whereby, using inks and specific chemical processes on a photograph edited especially for this reason, an image acquires a small relief [15].

Much more needs to be done; many ideas need to be generated, absorbed, and transformed into practice so that people with visual impairments can move and act without restrictions and thus bring out their full potential. It is necessary to listen to them and learn/understand their needs to meet them.

What is proposed in this research is precisely to seek parameters for using 3D printing in such a way that the perception of the two-dimensional visual origin material facilitates its understanding. And with the diffusion of these parameters, it is believed that there will be an increase in the flow of sharing visual information originally intended for people with disabilities, which will enhance understanding.

## 2. Tactile Content Manufacturing

Generally, materials for people with visual disabilities are constructed manually (using the attachment of layers of materials with different textures, thicknesses, and heights) or through manufacturing processes such as thermoforming and screen printing.

In thermoforming, the polymer sheet is pulled by vacuum against a matrix that has the desired shape, which is assumed by the sheet. This matrix can contain different levels of height, and thus the image becomes distinguishable both by shape and by differences in height [16].

Another commonly used technique is called thermal printing. In this process, a paper with a thermosensitive emulsion film (also known as Swell) receives the image from a photocopier, printer, or special pen. It is then subjected to infrared radiation promoted by a fusing machine. This process increases the height of the surface on the painted parts of the figure. Since the added height is the same for all parts, it is important to adapt the image to be 3D printed so that its content can be better understood by users given this limitation [16].

Screen printing presents a production process by applying ink to a matrix screen, which masks the passage of this ink to the desired locations for the formation of the figure. It is a long and more complicated process than thermal printing, but its results tend to be more durable, although it is more expensive in small productions [16].

Another method is stamping, in which the paper is pressed by two forms (which contain the positive and negative forms of the figure) for the formation of the image. Since this process allows for a preliminary impression, the generated page serves both blind and low-vision users. It requires a high initial investment for the creation of matrices, but allows for a high print run. This manufacturing method is used in the production of tactile books published in France [16].

From the point of view of artisanal processes, the most commonly found solutions are images constructed with different materials accessible in the surroundings. Often, the choice of material is based on the similarity or remembrance of a specific characteristic. There is a search to translate to touch a sensation similar to the visual impression that that material transmits; however, since the scale of the textures influences the perception performed, the same materials of the represented object can rarely be used [16].

## 3. Methodology

This study was conducted through experiments with production materials and visually impaired individuals in search of their perception of the produced elements.

The first experiments focused on finding a low-cost material to receive the 3D-printed “information”. Unlike the studies presented by Hasper, E. et al. [17], which used Corian (an acrylic polymer), MDF (medium-density fiberboard—a board made of medium-density wood particles), and HDP (high-density plastic—high-density plastic) under the machining of a CNC (computer numerical control subtractive manufacturing machine), here it was desirable that the variety be greater, encompassing economically and easily accessible materials, as well as being more suitable in terms of flexibility and adhesion, to serve as a base substrate for the rest of the tactile artifacts, aiming at durability under constant handling and archiving for later use.

With the materials, it would also be possible to discover how 3D printing characteristics would behave in terms of adhesion. The width and height of lines, their bending angles, and the influence of filled surfaces were investigated.

### 3.1. Material Selection

As presented in the previous section, this first part was dedicated to aspects of the interaction of printed material with the support media. The objective was to identify the low-cost medium with the best durability in relation to handling in order to reduce costs and speed up printing. To achieve this, different printing configurations were tested on sheets of different substrates and materials sold at affordable prices.

The experiments aimed to verify situations similar to real world use. For instance, suppose the tactile content is 3D printed as part of the pages of a children’s book. When the pages are manipulated by the children, they are usually deformed, and the tactile content must adhere to them. As media durability is a desired factor, to avoid having to manufacture new pieces every few uses, adhesion was chosen as a measurement parameter. The tests then involved the efforts suffered and measured as described in the following topics.

The materials were chosen based on their market availability and cost. Thus, financially accessible materials were sought in the local market. To define the value that would be used, a general survey of prices was conducted, and it was noticed that, sold per linear meter (with variations of one to two meters in width), the values in February 2020 showed a smooth increase up to R$ 7.00 and then rapidly jumped to over R$ 10.00, at the same time that the number of different samples decreased, and then made another jump to the twenties. Therefore, it was decided to define for the tests that the materials to be used should be found up to this first value, R$ 7.00/m^2^.

The materials found include synthetic and cellulose-based polymers (illustrated in Figure 1) and are listed as follows. Their thickness and price per square meter are described in Table 1.

Sulfite paper: also known as parchment, offset, or office paper, is one of the most common types of paper in offices. The samples used had densities of 75 g/m^2^ (Figure 1e) and later 180 g/m^2^ (Figure 1d).Pine cardboard: white cardboard, thought, solid, thin, and sold by weight. It has two faces with different textures and colors. One, lighter, is also rougher than the other (Figure 1f,g).Lisolene: found and known by this name in commerce, it was later discovered to be a polyethylene laminate, one of the simplest, most produced, and lowest-priced synthetic polymers. It has two faces with different textures: one is completely smooth (Figure 1a3), and the other has a pattern of small dots that do not reach the smooth side (Figure 1a2). It was acquired in two colors, transparent and opaque black (Figure 1a1), to analyze the hypothesis that color affects thermal energy exchange and, consequently, adhesion.TNT: non-woven fabric or not woven, are agglomerated fibers without the formation of a mesh (Figure 1c). In a later discovery, it was found that the fibers are fixed by a polypropylene resin, the second-most commonly produced synthetic polymer. The density of the acquired sample was 80 g/m^2^.EVA: ethylene vinyl acetate is sold as a sheet of soft and flexible synthetic polymer. It is usually specified by the thickness of the sheet. The one used was purchased as 2.5 mm thick (Figure 1b).Blackout/Transparent/Lamicel: sold under these different names, several samples were purchased only to discover later that it is the same material, polyvinyl chloride, abbreviated as PVC. In this case, it is presented as transparent, smooth, and flexible sheets that, when folded, form and maintain a rounded crease (Figure 1i).Vergé paper: similar to sulfite paper, but with visible internal horizontal lines resulting from the manufacturing process. The sample used had a density of 120 g/m^2^ (Figure 1h).

### 3.2. Tests Methodology

Regarding these materials, the objects were printed using the same spool of PLA, a biodegradable material that is cheaper and more widely used in the 3D printing technology employed, from the 3DLab brand.

There are some standardized norms for adhesion testing procedures, but none were found that focused on 3D printing on materials as diverse as those selected. Thus, it was decided to adapt and blend the norms from ASTM D4541-17 (2014) and ISO 11339-1 (2010) standards.

The ASTM standard defines procedures for evaluating binding forces between rigid metallic objects and presents different extraction methods.

The ISO method deals with film adhesion on surfaces. It was developed for metallic adhesives, but it claims that it can be applied to other flexible surfaces. Thus, it guides the experiments of Sanatgar, Campagne, and Nierstrasz [18], in which it was adapted for additive manufacturing of films on fabrics. In it, the printed film is separated from the base material by a tensioner that then records the force applied. In this movement, the printed layer is bent upwards when pulled, while the fabric curves downwards at the same time.

Concerning the reproducibility of the methodology, we adapted the test procedures. To solve this issue, a way of measuring which material would be the most adhesive was created using simple instruments. The basic idea was to bend the sample over a cylindrical object, which would indicate, for example, the resistance of the pages to the action of leafing through a book of tactile illustrations. The general steps are as follows:1.The material to be tested was cut into rectangles of the same size for each test set and labeled for discrimination.2.The rectangle was then fixed to the bottom left corner (from a top view) of the printer table (coordinate x = 0, y = 0) with adhesive tape.3.The desired shapes were printed onto the rectangles.4.After the printed sample was cooled, the rectangle was removed and fixed with adhesive tape to a cylinder with a radius of 22.85 mm.5.The set was rested on a flat and rigid surface so that the printed figure faced downwards.6.The cylinder was then rolled, causing the support material and, consequently, the printed material to roll around it.7.When the rotation was completed, the tape was released and the material was guided forward to allow it to continue rolling to the end without rolling over itself, as the circumference of the cylinder was smaller than the length of some samples.8.It was then removed from the cylinder, and the adhesion of the printed material was analyzed.9.If there was any detachment, the distance in a straight line from the detached end of the printed piece to the point where it was still fixed was measured (Figure 2).10.The material was then removed and fixed again to the cylinder, now on the opposite side to that used in the previous step, and the rolling and measurement process was repeated, thus using the opposite direction of rotation for a second measurement.11.The two measurements were recorded in a spreadsheet.

Sanatgar, Campagne, and Nierstrasz [18] state that the manufacturing parameters that influence adhesion are the extrusion temperature, printer bed temperature, print speed, 3D printing material, the material on which the object is printed, the dimensions of the printed object, and the type of infill. Assuming that all of these values can be maintained between prints, except for the support medium for 3D printing, the comparison of the recorded values may indicate the sample representing the most adhesive material.

Furthermore, by knowing the diameter of the cylinder on which the sample will be rolled (45.7 mm) and the thickness of the support medium, it is possible to determine the maximum angle of twist supported within that radius of curvature by applying a proportion of the circumference length of the set (radius of the cylinder plus the thickness of the tested material) relative to the measurement from the tip to the point of rupture.

This can be described by the following formula (please notice that all angles are measured in degrees):

$$\frac{360^{{}^{\circ}}}{\alpha }=\frac{C}{X}\to \alpha =\frac{360^{{}^{\circ}}X}{(d+2E)\pi }$$
where

*C* = circumference length of material set + cylinder,

α = desired angle (in degrees),

*X* = distance from tip to rupture location,

*d* = cylinder diameter,

*E* = thickness of the material being tested (from Table 1),

π = pi.

The experiments were made on a Prusa MK3S using a 0.4 mm nozzle and the same spool of PLA filament manufactured by 3D Labs. As temperatures below the glass transition temperature of the material to be extruded do not affect adhesion [18,19], the printer bed was heated to a temperature of 60 °C (used with PLA) to increase the stability of the surface molecules of the junction, improving contact and increasing Van der Waals forces [18].

The layers were printed with the nozzle at temperatures slightly above what is normally employed in PLA filaments: 210 °C for the first layer and 205 °C for the rest. The goal was to increase the thermal dynamics of the polymer macromolecules, which increases material penetration and diffusion over the surface [18]. For the same reason, the cooling fan for the printed material remained off during the construction of the first layer and gradually increased its rotation speed up to the fourth layer, thus ensuring slower cooling for greater adhesion [19].

In terms of adhesion between layers, 50 mm/s, when used in 0.2 mm layers, also generates the best results, considering efficiency, adhesion strength, print precision, and surface roughness [20]. Thus, the 3D printing speed of the structures was adjusted to these values. However, the height of the printed layer would not be compatible with all object heights tested. For this reason, it was decided to reduce it to 0.1 mm, a common multiple of the employed dimensions, so that the comparison between them would be homogeneous.

The first layer was also adjusted to 0.1 mm in height to increase the contact area with the tested material [21], and the print speed was set to 20 mm/s for this first layer. This number assumes that high speeds reduce the penetration of viscous polymers into tissues, while low speeds at high temperatures make the print brittle [18].

Then, the average thickness of the surface material on which the experiments would be printed was measured with a caliper at three different points and then added to the height of the first layer to be printed via slicing software (Z Offset option).

The general 3D printing settings are summarized in Table 2.

To ensure that the 3D printing always followed the same location in the material rectangle, the creation of digital parts included the generation of a frame with the same dimensions as the rectangle of the material to be cut. After aligning the object with the edges of the table in the slicer software, the frame object was deleted, leaving the objects to be printed in the desired position.

The objects were scaled in height to achieve the desired height for each test, and then the print file was generated. The printer was calibrated both in relation to the extrusion multiplier (to ensure the accuracy of the deposited filament thread width) and in relation to linearity compensation (to ensure that material deposition was uniform from start to finish) using the procedures described by the manufacturer.

The parts were printed directly on the substrates in three sets of tests: (a) adhesion, (b) angles, and (c) surfaces and shapes. The specific settings for each are shown in the next topics. First, they were printed once on 75 g/m^2^ Sulfite paper as a pre-test, in which all height options (as described in the next topic) were checked for the first three tests. Once the functionality and effectiveness of the process were tested, the experiments were initiated.

## 4. Tests

This section groups the tests performed into three major segments: adhesion, angles, surfaces, and shapes. All of them are described as follows.

### 4.1. Adhesion

The set of samples for the first test presented parallel lines with widths in multiples of the printer nozzle diameter (0.4 mm). This was done to not only discover which material would have the best adhesion but also to determine the influence of the width and height of the printed lines on how the sample would detach.

Each line was 5 mm away from the other, 20 mm from the edges towards their ends for fixation, and 10.4 mm from the side margins to both move away from them and to round the size of the material rectangle being tested, resulting in a width of 200 mm and a height of 40 mm.

The first line had a single perimeter. Due to the pressure from the nozzle, it spread to 0.45 mm in width. The others followed as multiples of the nozzle diameter due to the support that the deposited lines provided to the new ones. The extra 0.05 mm is shared between the lines, so that for more than one line, the width values are multiples of 0.4 mm. Thus, the other lines had a width of 0.8 mm (two perimeters), 1.2 mm (three perimeters), and 1.6 mm (four perimeters).

Figure 3 shows a top view of the constructive drawing of the adhesion experiment assembly. All lines were 160 mm in length. In this image, it is also possible to see the frame (outer rectangle) used for aligning the piece on the 3D printing bed in the slicer software and discarded before creating the print file.

Samples with the four lines were prepared for each material tested, varying the height of the lines: each group of four lines was printed with 0.1 mm, 0.2 mm, 0.3 mm, 0.4 mm, 0.5 mm, 0.7 mm, and 1 mm. This complete set of heights was printed and tested four times.

### 4.2. Angles

For the second test, the set of samples aimed to infer the influence of the angle in relation to twisting, and differs from the first experiment by having ten shorter lines (100 mm instead of 160 mm) of a single width (1.2 mm) and single height (0.7 mm) at different angles, between 0° and 90°, separated by 10° intervals, as shown in Figure 4, in which the dimensions of the sample rectangle of the tested material, 170 mm by 150 mm, and the margins of 20 mm and 10 mm can also be seen. Each set of angles was printed four times per tested material.

The 3D printing height of the lines was defined and fixed at a single value after pre-tests were carried out with all heights from the previous test. An intermediate value was chosen between that which allowed complete adhesion and that which caused complete detachment, as both situations would not allow differences between angles to be examined.

### 4.3. Surfaces and Shapes

To evaluate the influence of the filling pattern of the external surfaces (first layer and top printed layers), test 3 of this section placed three aligned geometric shapes, a diamond (50 mm × 30 mm), a circle (radius = 20 mm), and a square (side = 40 mm) in a rectangle of 60 mm by 200 mm, with 20 mm margins in the length direction and 10 mm in the width. The shapes were separated by 15 mm (Figure 5).

In the same way and for the same reasons as the previous test, the height was fixed at 0.5 mm, another intermediate value of adhesion after all had been pre-tested. The chosen surface patterns (top/bottom fill pattern, in the software) were rectilinear and concentric, as they were found to be an intersection in several slicing programs. The number of solid layers from the top and from the base were defined in order to create a solid object, without infill, and the contour lines (perimeters) were set to zero to not affect the behavior of the surfaces. The tests were carried out three times, for each material.

## 5. Results and Analysis

After removing the printer, cooling the assembly, and performing the detachment rolling and measuring stages, the values were recorded in an electronic spreadsheet. They were then subjected to numerical analysis and, subsequently, statistical tests. The first stage of this statistical analysis was to check the normality of the distribution of the data within each comparison. For this purpose, the Shapiro-Wilk [22] test was used.

In their absolute values, the variability of the data would be described by the standard deviation if the analyzed population followed a normal distribution and the sample size was not small (N > 50). Otherwise, the description would be given by the distance between the 25%, 50% (median), and 75% quartiles, a technique more appropriate for non-parametric or small sample cases where the mean used in calculations is strongly and easily influenced by the dispersion that may occur in the data [23].

As the test settings had independent variables, the Mann-Whitney (1947) test was used to compare two variables of the same configuration (in the case of comparing filling surfaces), and the Kruskal-Wallis [24] test was used when the same parameter had more than two variables (for the other aspects analyzed). In the latter case, the case-by-case comparative analysis, or parity test, was performed using Dunn’s post-hoc [25].

Thus, it was possible to verify whether the collected numbers presented significant differences or were statistically equal [26]. These statistical calculations were performed using the Jasp software [27], and the adopted significance probability was p≥0.05.

With all measurements cataloged in a spreadsheet, the results of the experiments are shown in six parts, related to findings about material adhesion, height, width, twist angle, shapes, and surfaces.

### 5.1. Material Adhesion

In this first test, 1344 measurements were made (six materials, four widths, seven heights, and four samples for each configuration, measured at two ends). As the measurements from both ends count as a single result for that sample, there are 672 samples.

Figure 6 shows a sample undergoing the procedure. On the left is the 3D printing process in the case of test 1. It is possible to observe the material fixed at the bottom left corner of the useful area of the 3D printing table, the purge line just below the sample, outside the print area, and the lines printed at a single height in the four different widths. On the right side of the image, the process of the first sample winding around the cylinder is shown, revealing the detachment of part of the lines, which would be later measured.

The materials behaved differently, and the reactions during the tests are reported below.

75 g/m^2^ sulfite paper

During the cooling of the samples, the difference in thermal characteristics of the materials, combined with the low thickness of the base material, caused the samples to curve upwards, with the axis perpendicular to the printed lines, and twist in inverse proportion to the height of the printed samples. This effect was found in other materials, but to a lesser extent. Figure 7 shows this occurrence in samples of 0.1 mm and 0.2 mm (at the top of the figure) against one of 0.7 mm.

During the pre-tests, attempts were made to prevent the curving in some samples in three ways: (a) letting the sample cool slowly on the printer table; (b) removing the sample and immediately storing it in a book with weights on top of it; and (c) letting the sample cool on a support that only touched its center, leaving its edges in the air under the action of gravity. None of the methods presented good results, but the second method slightly mitigated the effect on higher prints and the third method on lower prints.

In the second test, as it presented elements at angles between 0° and 90°, the curvature axis curved the paper at an average angle of the lines: 45° in the direction that cuts them (Figure 8).

In the third test of the surfaces, the curvature appeared even during 3D printing and more intensely, especially near the printed shapes, both in width and length of each geometric shape. This occurrence forced the resistance of the fixation to the printer table, which led to an increased use of adhesive tape to keep the material in place.

The curving was accompanied by a strong wrinkling on the sides of the filament deposition (Figure 9), which was already noticed in previous tests but to a lesser extent (Figure 7 and Figure 8). This effect created a texture around the printed elements and amplified the pulling of the sample’s sides.

Lisolene (polyethylene)

The heat from the nozzle melted and tore the Lisolene in several places while running over the surface. The printed lines came off very easily when the material was removed from the printer. Five attempts were made on both textures/sides of the Lisolene in both colors, always with the same result (Figure 10).

The reason for this result is due to the melting point of polyethylene, which varies between 105 °C and 125 °C depending on its density. Therefore, the experiment with this material proved to be unfeasible.

80 g/m^2^ TNT

The TNT was easy to fix to the 3D printing bed and performed very well under heat, without damage, tears or warping. However, the lines did not adhere to the material in any of the four attempts. And because it could not find adhesion, the filament remained stuck and accumulated in the nozzle.

It was during the 3D printing process that it was discovered that the fibers of the TNT are usually fixed with polypropylene resin, a synthetic polymer with high chemical resistance and therefore not compatible with most materials. Therefore, this material was also discarded.

Pine cardboard

One of the easiest materials to carry out the steps, the pine cardboard remained firm during 3D printing and received the material well. As it had sides with different textures, one lighter and rougher and another darker and smoother, they were considered different materials to verify if this roughness would interfere with adhesion.

In some of the samples of parallel lines with intermediate heights, there was a slight curving, replicating the effect seen on the sulfite paper, but in much smaller proportions and occurrences.

An easily noticeable effect was the interaction of the cardboard fibers with the printed material, especially on the darker and smoother side. When peeled off, it was possible to see them attached to the PLA, even making it difficult to identify the limit of detachment (Figure 11).

In several samples, part of the material was attached to the printed element, indicating that the adhesion to the printed part was greater than the adhesion between the layers of the material itself and was limited to this weaker bond. This was also verified by the constant need for renewal of adhesive tapes, which also shows the low adhesion between the layers of the cardboard itself, and it was at this point that most adhesion failures occurred.

On the right side of Figure 12, it can be seen that the cardboard material is well attached to the forms that were printed on the dark side of the sheet, while on the left side, where the samples of the light side are, there is less interaction between the fibers, and therefore the adhesion with the printed material is not as strong.

EVA

This was the only material that showed different measurements during the thickness check. Despite the fact that the purchased specimen was sold under the nominal dimension of 2.5 mm, the plate showed a random variation between 2.5 mm and 2.7 mm in various locations of the rectangles of the different samples. Thus, an average of 2.6 mm was chosen for the print configuration.

Due to the height, some modifications needed to be made to the 3D printing method so that the machine could start it without colliding with the edge of the material. After some tests, the solution found with the best result was to pause the print at the first millimeter of displacement of the initial purge line of the 3D printing material (PLA) (which caused the printer to automatically move the extrusion block away from the location and turn off its heating to avoid carbonizing the filament), then fix the sample on the lower right side of the table, and then resume the print so that the extruder continues the purge line (it automatically resumes the previously configured temperature and moves the extruder to the point where the pause occurred) and continues with the 3D printing work.

During 3D printing, there was contraction of the material due to the heat emitted by the extruder nozzle. This contraction generated grooves along the path taken by the nozzle during the deposition of the first layers (Figure 13). As a result, material printed in samples of 0.1 mm and 0.2 mm in height did not show up above the average surface of the EVA.

An attempt to reduce the groove by increasing the distance from the nozzle to the material (Z Offset) by 0.1 mm resulted in a complete lack of adhesion of the printed elements. So this idea was discarded, and the previous 3D printing method was returned to.

Lamicel (PVC)

Like polyethylene, this material also experienced melting in some areas traversed by the print head. But, unlike what happened there, the adhesion of PLA was perceived, especially in the PVC melting locations. However, these regions were characterized by hardened and sharp fractal deformations that would certainly interfere with the perception of the drawn material (Figure 14, left side).

Therefore, it was decided to change the pre-defined settings in the methodology to eliminate the problem. Thus, the temperature of the print head was reduced and the 3D printing speed was doubled, so that it would spend less time on each point of the material. The fan that remains directed towards the material deposition point from the first layer was also activated, aiming to cool the region and, consequently, prevent melting. These new settings are presented in Table 3.

These new settings did not completely eliminate melting, although they reduced its occurrence (Figure 14, right side). However, they greatly affected adhesion and some lines did not stick to the material. Others came off when the assembly was removed from the printer. The ones that stayed did not withstand rolling. Therefore, this material was also discarded from the experiment.

Vergé paper 120 g/m^2^

Similar to what happened with sulfite paper, vergé paper also curved during cooling in the same axes, but to a lesser extent than the sulfite and greater than the pinewood cardboard. Except for the post-cooling state, it behaved rigidly, becoming one of the materials with the fastest assembly and execution and the lowest number of failures among all tested.

Sulfite paper 180 g/m^2^

Given the adhesion results of the 75 g/m^2^ sulfite paper and its wrinkling, in addition to the good results of the 120 g/m^2^ vergé paper, the 180 g/m^2^ sulfite paper was added to the tests. The hypothesis was that it would maintain the adhesion of the 75 g/m^2^ sulfite with rigidity equal to or greater than that of the 120 g/m^2^ vergé paper.

During 3D printing, it was more rigid than the vergé paper, with less curvature, practically no wrinkling, and the smoothest surface (both in terms of roughness and non-wrinkling) among all materials. It was also one of the easiest procedures to perform.

Behaviors and Performance

A total of 672 samples passed through the adhesion test settings 1112 of each material (with variations of seven heights and four widths in four printed specimens of each configuration).

The 75 g/m^2^ sulfite paper showed the best adhesion, with a median of 100%. The 25th percentile indicates that 75% of the results showed a length above 96.516% still adhering after rolling (Figure 15). However, as previously mentioned, this material also accumulated the highest surface deformations. The 120 g/m^2^ laid paper also falls into this higher adhesion group, equaling the high median of the 75 g/m^2^ sulfite results (100%). But its 25th percentile was lower than that of the other material (75.609%), and its wrinkling was less intense.

The dark side of the pine cardboard and the 180 g/m^2^ sulfite paper come in a second group of materials with good adhesion. The dark side of the pine did not show wrinkling due to its greater thickness (0.75 mm). It obtained a 25th percentile of 80.719%, indicating little dispersed results. It also obtained a slightly lower median than the previously described materials, which indicates that half of the samples maintained more than 97.063% of their length adhering to the base material. The 180 g/m^2^ sulfite paper presented an intermediate result with adhesion close to the highest (median of 99.313%) with very low wrinkling and distortions. However, half of the samples that were below the median showed dispersed results, leading the 25th percentile to zero under the test configuration conditions.

In the third group of results, the light side of the pine cardboard and the EVA are found. Both showed good adhesion results only at the lowest tested heights. The medians are among the lowest: 91.969% for pine and 82.719% for EVA. Both also generated a 25th percentile of 0% and a 75th percentile of 100%, showing great amplitude in their results.

The Shapiro-Wilk test [22] indicated that the adhesion results were presented under a non-parametric distribution (*p* < 0.001 for all of them). In the parity test after the Kruskal-Wallis test [24], the distribution of the mentioned groups was confirmed. Three pairs of materials were shown to be equal: (1) the adhesion of the 75 g/m^2^ sulfite paper and the laid paper (*p* = 0.280); (2) the results of the 180 g/m^2^ sulfite paper and the dark side of the pine cardboard (*p* = 0.469); and (3) the numbers of the light side of the pine cardboard and the EVA (*p* = 0.419). The remaining parities were significantly different from each other (*p* < 0.001) (Figure 15).

### 5.2. Height

Each of the seven printed line heights was tested 96 times in test 1, using the different materials and widths described in the methodology. At a height of 0.1 mm, there was no detachment. The lines remained completely fixed after the two winding processes in all samples, regardless of the material or width of the lines.

The 0.2 mm and 0.3 mm heights presented minimal detachment, closing the three quartiles (25th percentile, median or 50th percentile, and 75th percentile) with 100% adhesion for both heights.

The printed height of 0.4 mm was shown to be the limit of these high adhesion values; the median was established at 98.344% of the length of the lines, remaining fixed to the materials. The 25th percentile indicates low dispersion by positioning 75% of the samples above 90.953% of the adhesion of the length of the lines.

From then on, there was a rapid decrease in values: the 0.5 mm height presented a median of 92.844%, which drops to only 3.41% when the height reaches 0.7 mm. The 25% and 75% percentiles increase their distance from the median, indicating a more varied behavior when winding, ranging from 47.203% and 99.078% at 0.5 mm to 0% and 82.75% at the height of 0.7 mm.

The printed height of 1 mm provided the greatest resistance to curvature, with some portion of the printed element fixed in a few occurrences, leading to three quartiles of 0%.

Figure 16 presents this information. It is possible to observe the drastic decrease in the median between heights of 0.5 mm and 0.7 mm and the increase in dispersion at this point.

When converted into angles, it can be seen that the printed heights of 0.1 mm and 0.2 mm supported 393.08° of torsion at the median. This value considers the influence of variations in the thickness of the base materials on the calculation of circumference and, consequently, on torsion angles. Thus, the length of 160 mm of the lines corresponds to values between 360.21° in the thickest material, EVA, and 399.45° in the thinnest material, 75 g/m^2^ sulfite, passing through the 388.45° of pine cardboard and the 397.71° of vergé paper and 180 g/m^2^ sulfite.

As the height of the prints increased, the medians decreased: 388.45° at 0.3 mm; 382.22° at 0.4 mm; 360.65° at 0.5 mm; 12.84° of curvature at the height of 0.7 mm, showing that half of the samples supported bending up to this angle; and the three quartiles at 0° for the printed height of 1 mm (bottom part of Figure 16).

The sample distribution proved to be non-parametric (*p* < 0.001 for all, except for the height of 0.1 mm, which obtained the same result in all samples).

By checking the statistical significance of the differences between the results of each height through the Kruskal-Wallis test [24] and Dunn’s post-hoc test [25], it can be seen that there is similarity in the adhesion effectiveness results between the heights of 0.1 mm and 0.2 mm. The adhesion results at the other heights are significantly different (Figure 16).

### 5.3. Width

Each width of the lines underwent 168 tests among the different materials and heights. The two best conditions were presented at intermediate widths. 0.8 mm and 1.2 mm had median adhesion of 100% and 99.5%, respectively. The width of 0.45 mm showed the lowest result, 98.344%, while the width of 1.6 mm had a slightly higher average, 96.969%. In these last two cases, there was also a great dispersion of results, with the 25th percentile of the width of 0.45 mm reaching 0% and that of the width of 1.6 mm reaching 12.484% (Figure 17).

Mirroring these numbers, the median angles of adhesion were 388.45° when isolating the samples of 0.8 mm width and 381.85° for the width of 1.2 mm.

The width of 0.45 mm achieved a median of 363.94°, while the width of 1.6 mm had half of its results above 360.21°. The normality test indicated non-parametric samples. After the Kruskal-Wallis test [24], the post hoc Dunn test [25] showed a distinction between the results of widths of 0.45 mm and 0.8 mm (*p* = 0.019) and between 0.8 mm and 1.6 mm (*p* = 0.024). On the other hand, the closest similarities were found between 0.45 mm and 1.6 mm (*p* = 0.464) and between 0.8 mm and 1.2 mm (*p* = 0.316). There was a close similarity between 0.45 mm and 1.2 mm (*p* = 0.056) and between 1.2 mm and 1.6 mm (*p* = 0.067).

Comparing the results of the widths with the heights of the printed elements, it was observed that the thinner width (0.45 mm) mainly affected the heights of 0.4 mm and 0.5 mm, but it behaved similarly to the others in other heights. The 75th percentile for a height of 0.7 mm never reached 100% in any width, while at a height of 1 mm, the maximum value never reached this number, remaining at 96.06% and showing almost zero adhesion in the vast majority of its results. Figure 18 illustrates the evolution of these data, grouping the seven heights into four categories based on the widths.

### 5.4. Twist Angle

It was expected that the adhesion of lines would decrease linearly and constantly as the twist angle increased from 0° to 90°, with lines perpendicular to the twist experiencing the greatest opposing force. However, what was observed was a heterogeneous loss of adhesion from 0° to 30°, followed by a rapid increase up to 70°, after which it stabilized.

In the upper graph of Figure 19, it can be seen how the median lines almost form a steep straight line from 40° to 60°. In addition, the behavior was quite erratic at low angles (high perpendicularity), with medians ranging from 44.8% at 0° to 18.45% adhesion at 30°.

In other words, the printed lines that were closer to perpendicularity with respect to the twist exhibited lower adhesion with greater dispersion of their results. This was easily noticeable up to 40°. From 60° to 90°, the adhesion results were much more concentrated, as can be seen by the proximity of the 25th and 75th quartiles. The median increased up to 80° and then remained constant.

Statistically, the sampling appears to be non-parametric (*p* < 0.003) and the Dunn post hoc test [25] on the Kruskal-Wallis test [24] indicates that the adherence between angles configures two blocks: between 0° and 50°, differences in angle values are not significant in all possible combinations of angles (0.089 < *p* < 0.449), but become significant in combinations between these angles and any others above 50° (*p* < 0.001 to *p* = 0.010).

The same happens with angles between 70° and 90°, where there are significant similarities (0.262 < *p* < 0.409), and then there are significant differences again in relation to angles below them (*p* < 0.001 to *p* = 0.044). Figure 19 illustrates these relationships and allows the visualization of the two blocks of relationships that characterize adherence in this study and the isolation of the 60° angle.

The third test evaluated two surface-filling shapes and three solid geometric shapes in search of their influences. There were 216 measurements in 108 samples.

Due to the larger contact area, all materials showed greater adherence in this test than in the previous ones, which only counted lines, even on materials with lower adherence results, such as EVA. However, with a lower value for the length of the analyzed shapes and a higher value for the contact area, it is not possible to directly compare the adherence of these samples with those of the previous tests, but it is possible to observe the different reactions between the variables presented and tested here.

### 5.5. Shapes

After analyzing the 36 samples for each geometric shape, the square was found to be different from the other shapes: half of the samples maintained more than 86.125% of the fixed length after the winding stage. The second shape with the best result was the diamond (median at 41.5%) and, closer to it, the circle (median of 31.5%) (Figure 20).

The diamond’s highest result was 80.6% of the length remaining adherent, while the others reached 100% twice (circle) and seven times (square). The circle was the only shape with one of the quartiles at 0%.

Statistically, the sample was non-parametric, and the Kruskal-Wallis test [24] with Dunn’s parity test [25] indicated that there is significance in the proximity between the values of the diamond and the circle (*p* = 0.382). However, both generated significantly different results compared with the square (*p* < 0.003).

### 5.6. Surfaces

The results of 3D printing with different bottom and top surfaces, varying between rectilinear and concentric, showed similar overall average results: 49.94% and 43.12%, respectively, of the 54 samples for each surface. The medians were slightly further apart: 50.3% using rectilinear and 36.4% using concentric (horizontal lines in the centers of the rectangles in Figure 21. The increase in difference was affected by the results of the diamond shape, which had slightly better adhesion when printed with a rectilinear surface. This is evident in Figure 22, which compares the quartiles of both surfaces for each geometric shape. In addition to this gain with the diamond shape, it is possible to see that, despite having slightly lower medians in the circle and square, the concentric surface obtained more concentrated results than the rectilinear surface.

As there were only two variables, the statistical test used to verify differences was the Mann-Whitney test [28]. With a *p*-value of 0.317, it can be affirmed that the variations in results are not sufficient to declare a difference between the types of surface filling.

Cellulose-based materials performed well in adhering to PLA, especially those with higher densities. Printed line heights were found to be the main influencers of adhesion, which rapidly drops when varying from 0.5 mm to 0.7 mm when constructed with widths between 0.45 mm and 1.6 mm.

The widths had very close results, with a slight advantage for 0.8 mm and 1.2 mm, respectively, two and three perimeters in a 0.4 mm diameter printing nozzle.

In the variation of angles in relation to the torsion axis, those between 0° and 20° in relation to the torsion axis stood out. From perpendicularity to 50° in relation to this same axis, there were no significant differences in the low adhesion results.

The pattern of filling of the top and bottom surfaces did not significantly affect adhesion, but the square geometric shape obtained better results than the diamond and the circle.

With these values located for the tested variables, some of them were selected for application with volunteers, the real validators of the practical knowledge of this research. This evaluation is described in the next section.

## 6. Conclusions

3D printing on a base made of a different material than that used in printing brings the benefits of great time and material savings. In some of the experiments conducted here, savings reached 4050% in terms of time and 2914.29% in terms of material. These values vary from part to part and are inversely proportional to the percentage of the base area covered by useful information.

However, the adhesion of the printed material to the base depends on the choice of the base material. Chemical affinity and thermal stability at high temperatures point to the choice of cellulose-based materials with grammages above 120 g/m^2^ as the ideal solution.

However, the design of tactile content also depends heavily on the settings chosen for the construction and 3D printing of the information. Generally, the adhesion of contour lines is limited to certain 3D printing heights. The information would not withstand, for example, a crease fold. However, at a height of 0.1 mm, adhesion was sufficiently high for 96 straight lines of 160 mm to withstand complete detachment after two complete windings on a cylinder, perpendicular to its axis: 360.21° to 399.45°, depending on the thickness of the tested material, over 22.85 mm of radius. With 0.2 mm, the results showed no significant differences. Good results were maintained up to a height of 0.4 mm. Lines with greater heights lead to objects with low adhesion to the base and may come loose during user handling.

Several ideas emerged during the construction of this research. They are listed in the following paragraphs.

The first idea is the exact determination of the influence of the temperature of the bed and air humidity on the production of tactile content printed on base materials. It is known that an increase in the temperature of the bed enhances adhesion, but with the isolation provided by the material, it would be interesting to know how much of the temperature reaches the surface where the polymer filament will be deposited. Additionally, it would be interesting to know if and how much air humidity can affect adhesion in cellulose-based materials.

Another intriguing aspect is the possibilities that 3D printing using a flexible material filament can generate. If the adhesion to the base materials is good, it would offer very low resistance to bending of the content and thus allow for greater durability.

On the other hand, it is a material that is difficult to apply, given the ease with which it clogs the print nozzles since, being flexible, the same pressure cannot be applied for material extrusion or pulling for removal from the machine. For these reasons, retraction during 3D printing needs to be turned off, and the deposition needs to happen extremely slowly, which affects the time and consequently the production costs.

## Figures and Tables

**Figure 1 polymers-15-02180-f001:**
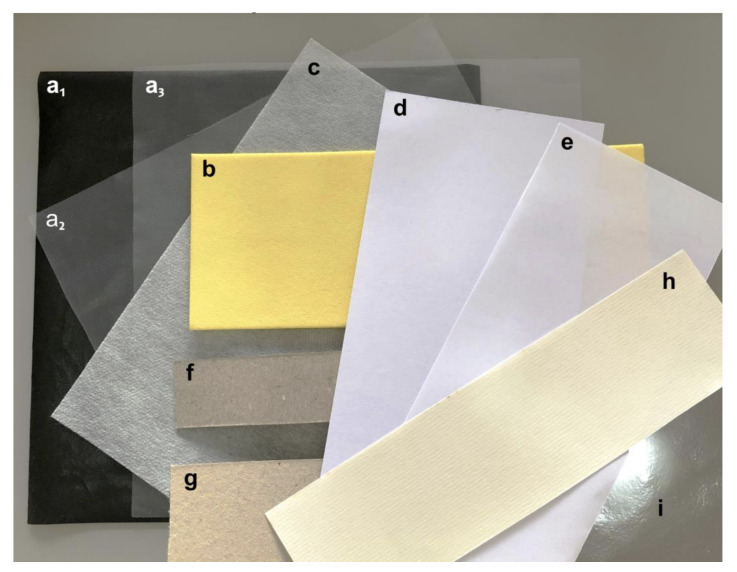
Materials found: Lisolene (**a1**,**a2**,**a3**), EVA (**b**), TNT (**c**), Sulfite paper (**d**,**e**), Pine cardboard (**f**,**g**), Vergé paper (**h**), Blackout/Transparent/Lamicel (**i**).

**Figure 2 polymers-15-02180-f002:**
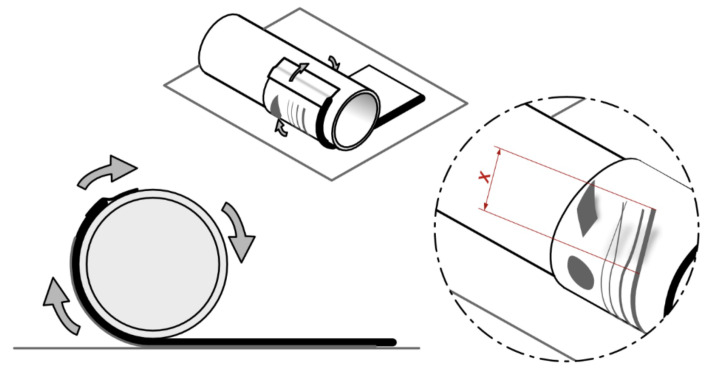
Adherence test. This figure illustrates the rectangle containing the 3D printed material attached to the cylinder and how it behaved on the adherence tests. The highlighted view shows some of the 3D printed content (lines and shapes) detached from the base surface due to the rotation performed. X corresponds to the distance in a straight line from the detached end of the printed piece to the point where it was still fixed.

**Figure 3 polymers-15-02180-f003:**
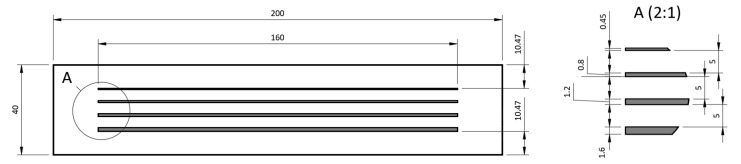
Top view of the construction design of the adhesion experiment assembly.

**Figure 4 polymers-15-02180-f004:**
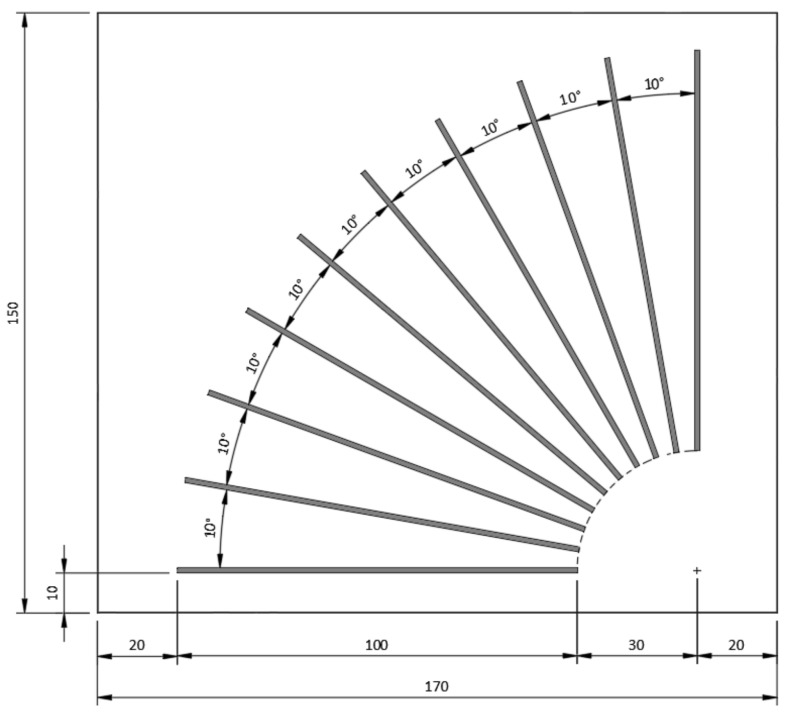
Top view of the experimental setup with lines at different angles.

**Figure 5 polymers-15-02180-f005:**
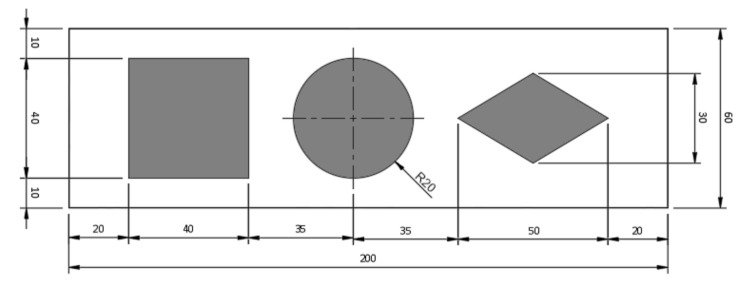
Top view of the experimental sample’s construction design with surfaces and shapes.

**Figure 6 polymers-15-02180-f006:**
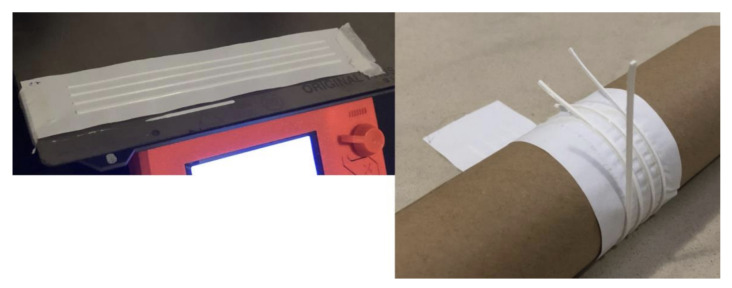
3D Printing and winding process of the adhesion test.

**Figure 7 polymers-15-02180-f007:**
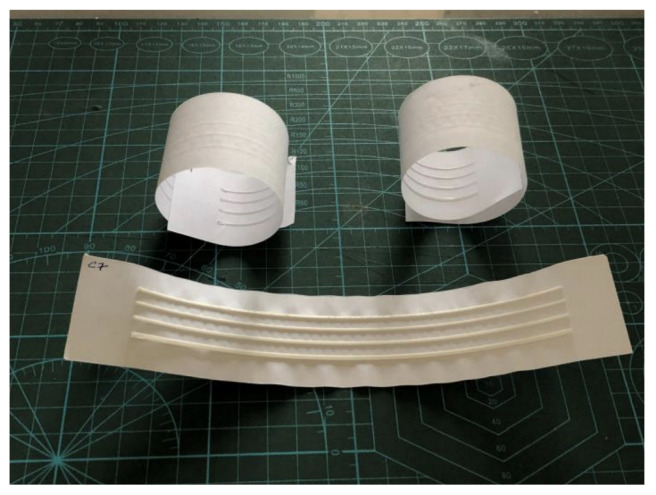
Curvature effect observed in the printed samples of the first test (using 75 g/m^2^ Sulfite paper as substrate).

**Figure 8 polymers-15-02180-f008:**
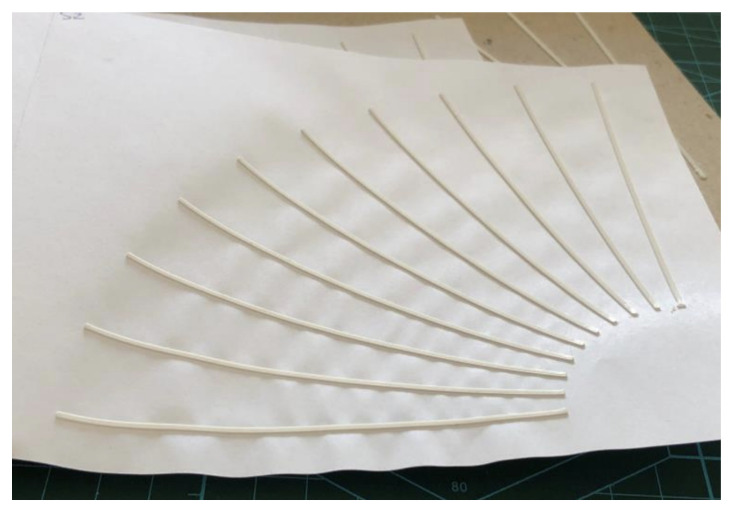
Curvature effect on the second test (using 75 g/m^2^ Sulfite paper as substrate).

**Figure 9 polymers-15-02180-f009:**
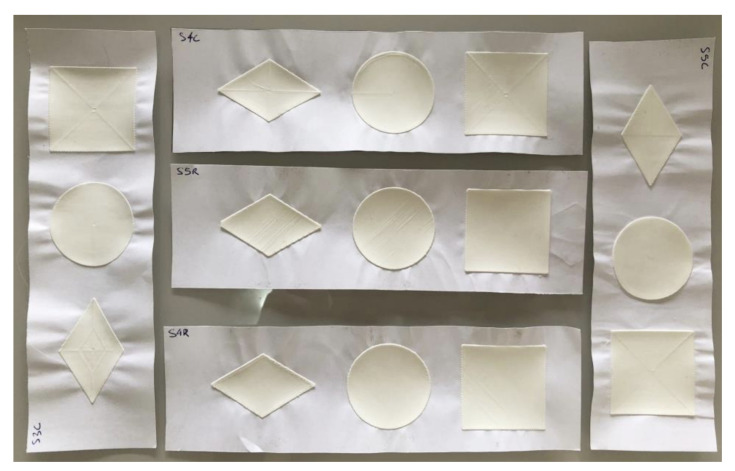
Wrinkling of the material on the edges of the printed shapes (using 75 g/m^2^ Sulfite paper as substrate).

**Figure 10 polymers-15-02180-f010:**
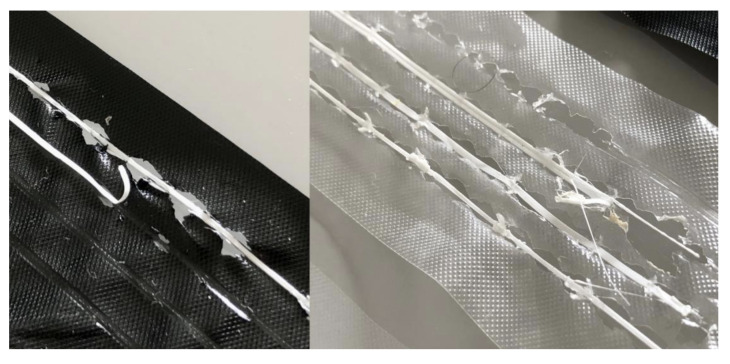
Effect of heat on Lisolene.

**Figure 11 polymers-15-02180-f011:**
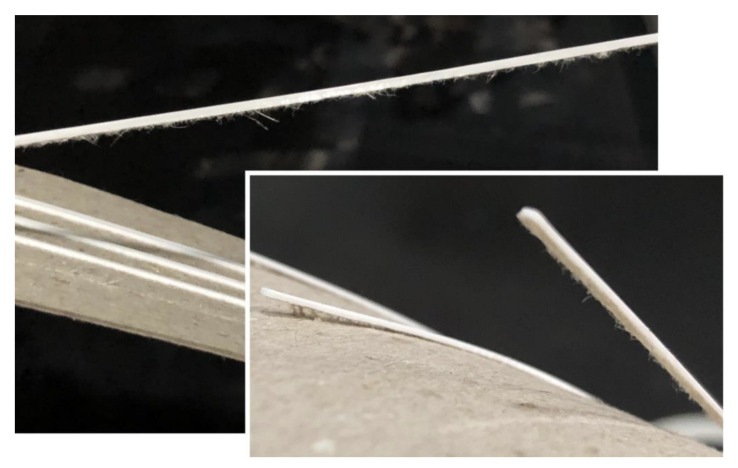
Interaction of PLA with the fibers of pine cardboard.

**Figure 12 polymers-15-02180-f012:**
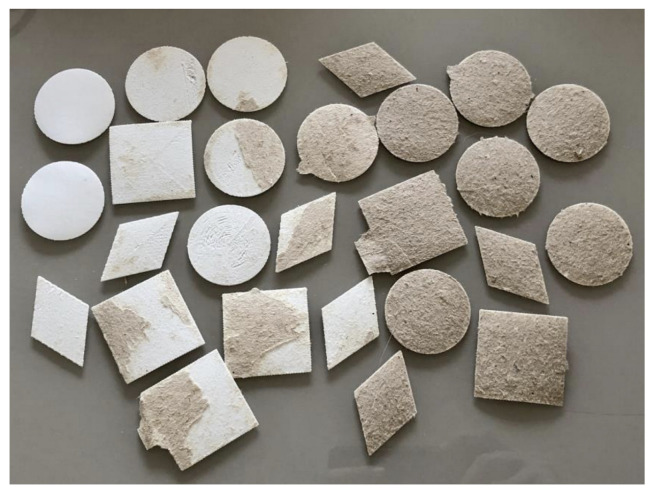
Cardboard residue during the detachment of the third test.

**Figure 13 polymers-15-02180-f013:**
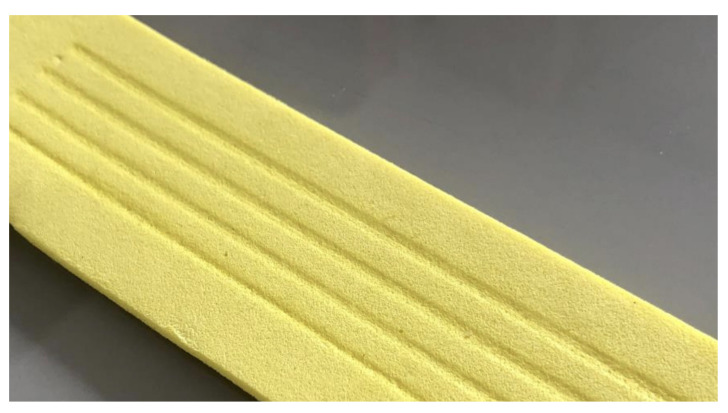
Grooves engraved in the EVA by the heat of the nozzle.

**Figure 14 polymers-15-02180-f014:**
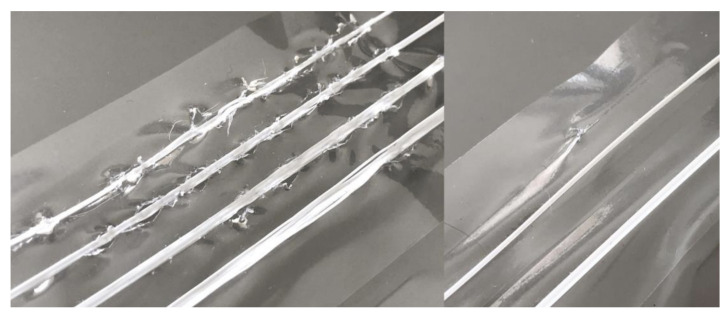
Result of attempts to print on PVC.

**Figure 15 polymers-15-02180-f015:**
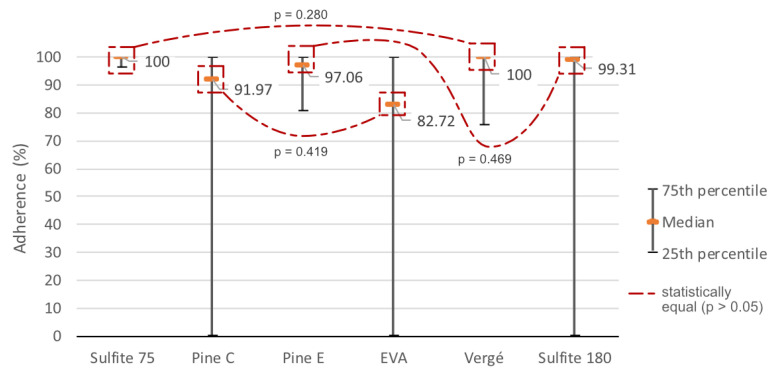
Material Adhesion (%).

**Figure 16 polymers-15-02180-f016:**
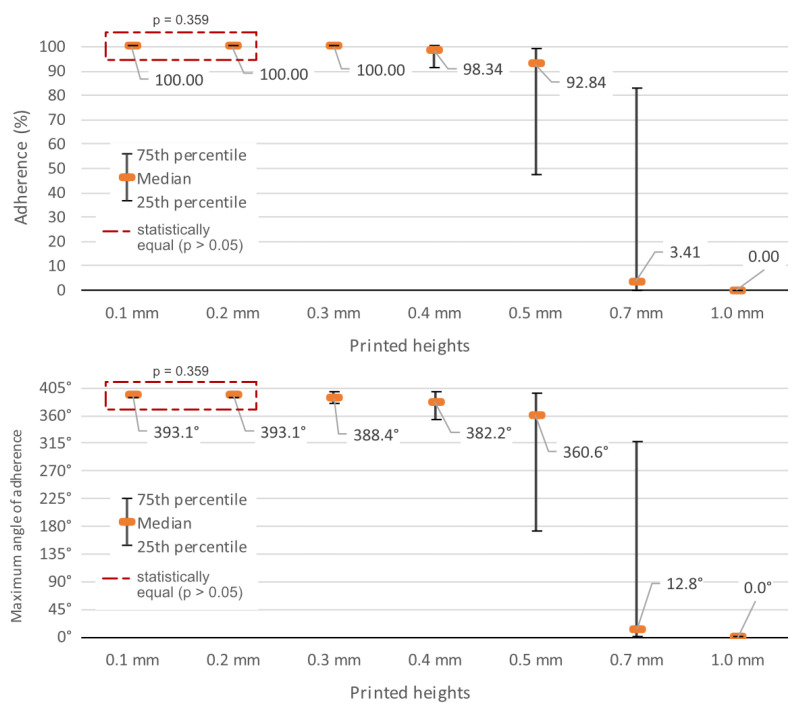
Percentage and angle of adhesion of the samples by printed height.

**Figure 17 polymers-15-02180-f017:**
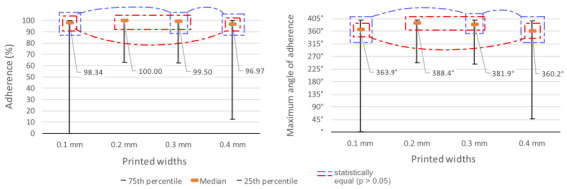
Percentage and angles of adhesion results by width.

**Figure 18 polymers-15-02180-f018:**
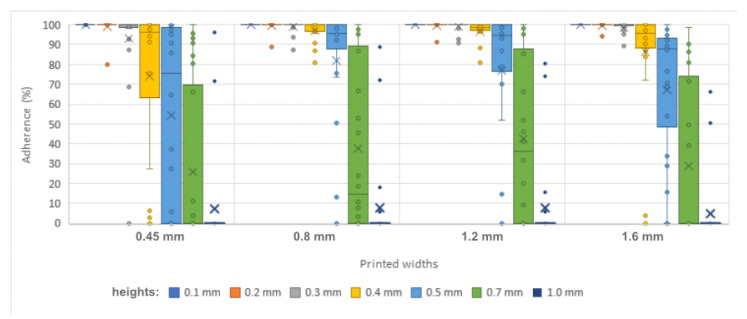
Relationship between adhesion and the seven heights and four widths tested. From left to right: width of 0.45 mm (1), 0.8 mm (2), 1.2 mm (3), and 1.6 mm (4).

**Figure 19 polymers-15-02180-f019:**
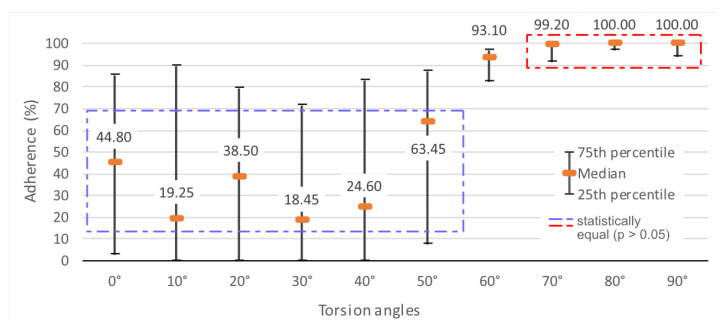
Percentage and angles of adhesion of lines by twist angle.

**Figure 20 polymers-15-02180-f020:**
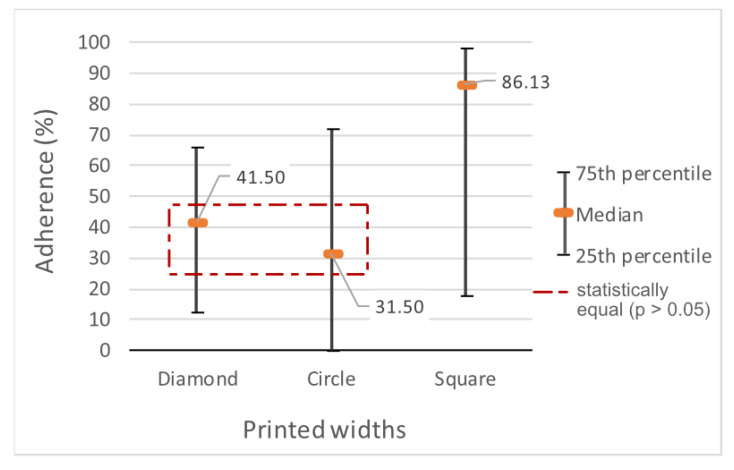
Percentage of the length of the shapes that remained adherent.

**Figure 21 polymers-15-02180-f021:**
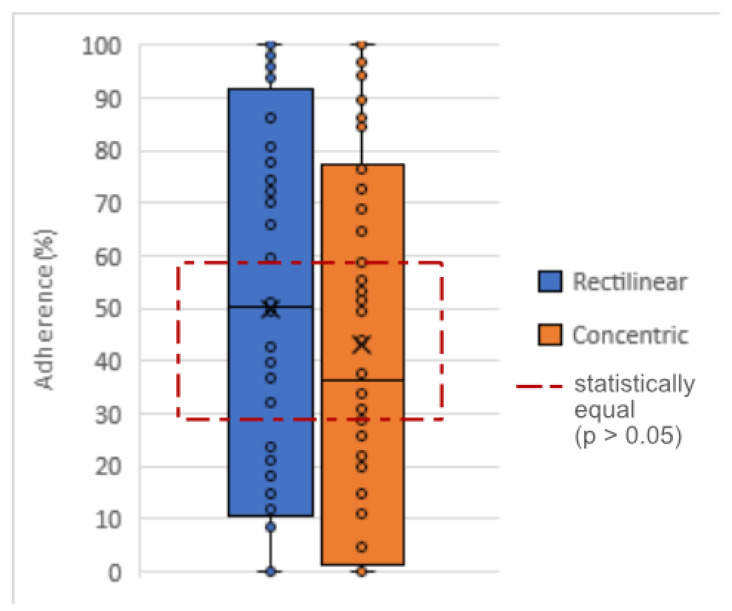
Distribution of adhesion of samples for different surfaces.

**Figure 22 polymers-15-02180-f022:**
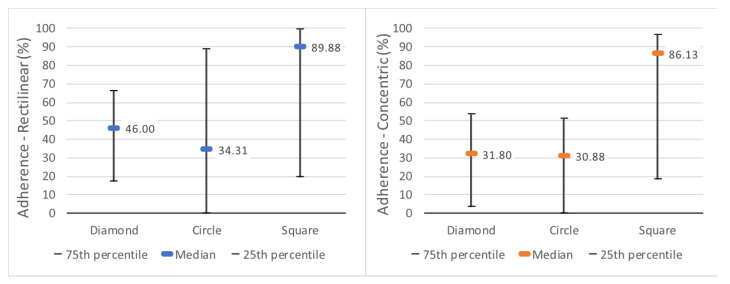
Differences in adhesion between shapes by surface type. Rectilinear on the left, Concentric on the right.

**Table 1 polymers-15-02180-t001:** Average thicknesses and prices of the materials used.

Material	Thickness (mm)	Price per Square Meter (R$)
Sulfite paper 75 g/m^2^	0.1	0.74
Lisolene (polyethylene)	0.1	1.33
TNT 80 g/m^2^	0.43	3
Cardboard	0.75	2.86
EVA (ethylene vinyl acetate)	2.6	7
Lamicel (PVC)	0.1	2.5
Verge paper 120 g/m^2^	0.2	3.7
Sulfite paper 180 g/m^2^	0.2	0.42

**Table 2 polymers-15-02180-t002:** 3D Printing settings for the adhesion test parts.

Parameter	Element
Height of the first layer	0.1 mm
Height of the remaining layers	0.1 mm
Perimeter lines	2
Lines on the skirt	0
Temperature of the first layer	210 °C
Temperature of the remaining layers	205 °C
Temperature of the bed	60 °C
Cooling fan speed for printed filament (min./max.) (Fan speed)	0%/50%
Number of layers with the fan turned off (Disable fan for the first)	1
Layer at which the fan will reach maximum speed (Full fan speed at layer)	4
Initial 3D printing height (Z-Offset):	Material thickness (see Table 1)
Contour filament 3D printing speed (perimeters)	50 mm/s
Solid infill 3D printing speed	50 mm/s
Top solid infill layer 3D printing speed	50 mm/s
Inter-filament gap 3D printing speed	50 mm/s

**Table 3 polymers-15-02180-t003:** Settings used in the PVC test.

Parameter	Element
Temperature of the first layer	200 °C
Temperature of the other layers	200 °C
Cooling fan speed for printed filament (min./max.) (Fan speed)	50%/100%
Number of layers with fan turned off (Disable fan for the first)	0
Layer at which the fan will reach maximum speed (Full fan speed at layer)	2
Filament contour 3D printing speed (perimeters)	100 mm/s
Solid infill 3D printing speed	100 mm/s
Top solid infill 3D printing speed	100 mm/s
Interfilament filling speed (gap)	100 mm/s

## Data Availability

The data presented in this study are available on request from the corresponding author. The data are not publicly available due to ethical restrictions.

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
