# Peer review of "Towards the Effectiveness of 3D Printing on Tactile Content Creation for Visually Impaired Users"

_polymers, 2023, doi:10.3390/polym15092180_

Round 1
Reviewer 1 Report
Dear Authors,
in your interesting manuscript, the following points should be added/changed to further improve it:
- Abstract: Please write m² with superscripted "2". It is not reasonable to give a bain in material consumption with 6 significant digits. The width and performance heights are not defined and thus not understandable. Which volunteers' activities? Which angles? Why are there two different infill density ranges?
- line 25: This sentence doesn't make sense.
- What is "beyond the world"?
- line 84: "the use of Marc Dillon"?
- Fig. 1: Did you take these photographs yourself? If not, a copyright notice with the received permission is necessary.
- line 87: What is meant here?
- Section 3 is highly unclear. There are a lot of wordings which don't say anything, like "a methodology of project and processes", "inductive approach", "solved in sequence" (how else?), "material dimension" (material or dimension?), "functions indicative of the analysis", ..., "finding a low-cost material to receive the information" (which?), etc. Please write down in a clear, concise way what you did, without adding a lot of unnneccessary phrases. This is also valid for the sub-chapters 3.1 ff.
- Fig. 2 is unclear. What is, especially, shown in the inset?
- Eq. in line 216: 360° misses a unit. A is not a common abbreviation for an angle, why not just alpha? M is not correctly defined - a physical parameter cannot be "measurement", but is apparently a length here. Phi, on the other hand, is normally an angle, not a diameter (why not just d?). Which base material?
- After reading the whole section 3, I read a lot ot things which are useless (or at least their necessity is not yet clear), such as G-code elements, but on the other hand I have no idea what you really did - I didn't even find the printer, not to talk about the filament or the printed samples, infill density or pattern, and especially the setting of the z-distance which is most important for the adhesion. And what about the substrates you printed on? Normally for a simple 2D printing process, all information about sample preparation fits on max. half a page, and maybe another half page is necessary for the description of the tests.
- line 287: You can definitely not give dimensions like 10.475 mm with 5 significant digits; the printer is not so exact.
- "Due to the pressure from the nozzle, it was resized" - by you or in an undesired way? If a second line is added, why are 0.45 mm of the first line + 0.4 mm of the second line = 0.8 mm?
- Fig. 3: The numbers are not readable.
- line 347: Which 2040 measurements?
- Fig. 7 finally shows the adhesion test. So why was this roll diameter chosen? And what was the reason not to use any of the common adhesion tests which all the groups working on 3D printing on textile fabrics use?
- Fig. 10: Which substrate materials are visible here?
- Fig. 13: Where is the border between both sides of the substrate?
- line 480 ff: How were the substrate heigths measured?
- Fig. 16: Please use English labels. The right axis needs units. It is not possible to connect data if the x-axis is not linearly, logarithmically or similarly spaced (there is simply nothing between EVA and Vergé etc.). What do the red dotted lines mean?
- Fig. 17-23: Again, please use English labels and, correspondingly, decimal points. And if you want to connect the dots, it is necessary to space the x-axis equidistantly. What to red boxes mean?
- line 619-620: Are the standard deviations small enough to allow for giving angles with 5 significant digits?
Fig. 19: Please add the missing units. Why are the four parts not directly labeled with the widths?
- Fig. 23: Connections between points are not allowed.
- It is not possible to have a sub-section 6.1 without 6.2.
- Author contributions are missing.
Generally, your work may be interesting, but is is urgently necessary to heavily rewrite the whole manuscript so that the normal scientist has a chance to understand it. I assume you can skip 1/4-1/3 of the text. Use more tables, shorter sentences, write clearly to the point. This is not philosophy, this is materials science.
unneccessarily elongated text
Author Response
Dear reviewer,
Thank you for your careful review of our paper. We appreciate your feedback and comments. We have addressed all of your concerns and have made significant revisions to improve the quality of our paper.
We have addressed the issues you raised concerning paper understanding and organization. We have also re-evaluated our writing using Grammarly. We believe that these revisions have significantly strengthened our paper and have addressed all of the concerns that you raised.
Once again, we appreciate the time and effort you invested in reviewing our paper, and we hope you will find the revised version suitable for publication. Thank you for your consideration. Please let us know if further modifications are necessary. We will try our best to address all of them.
The responses are highlighted in blue. The new PDF of the manuscript highlights the modifications in red.
Sincerely,
João Marcelo Teixeira
In your interesting manuscript, the following points should be added/changed to further improve it:
- Abstract:
Please write m² with superscripted "2".
Thank you for pointing that out. The superscript was fixed in both occasions.
It is not reasonable to give a bain in material consumption with 6 significant digits.
This part of the text was rewritten to simplify its understanding and show the gain accordingly.
The width and performance heights are not defined and thus not understandable.
We have modified the abstract to include a brief explanation regarding the line widths and material adherence.
Which volunteers' activities?
We have added a brief explanation in the abstract regarding the visually impaired activities used to measure different 3D printing parameters focused on adherence. We hope that it became clearer now.
Which angles?
The angles regard bending the surface that holds the 3d printed content. We observed that if lines are 3D printed between 0 to 20 degrees of the bending direction, the adherence is kept. Higher values lead to poor adherence on the surface. We have clarified the explanation regarding the bending angles on the abstract, thank you for pointing that out.
Why are there two different infill density ranges?
We performed tests regarding texture with two infill patterns. In the first case, we noticed that by using a single infill pattern, let's say, the rectilinear one, visually impaired users were able to discriminate the tactile surface up to 50% of infill. In the second case, when we combined two infill patterns, the discrimination increased and up to 90% of infill could be used. Thank you for asking for clarification about that topic in the abstract. We tried to make it more clear in the new version of the abstract.
- line 25: This sentence doesn't make sense.
Thank you for pointing that out. The sentence was replaced for better comprehension.
- What is "beyond the world"?
By "beyond the world" we meant the use of 3D printing by astronauts, in space. We have added a reference to make this sentence clearer, thanks for pointing that out.
- line 84: "the use of Marc Dillon"?
Thanks for pointing that out. The entire paragraph was rewritten for improved comprehension.
- Fig. 1: Did you take these photographs yourself? If not, a copyright notice with the received permission is necessary.
Thank you for noticing that. Figure 1 was removed together with its referencing paragraph. We believe this will not harm the paper understanding as the previous paragraph already contains references to such works.
- line 87: What is meant here?
This part of the text was rewritten to make it clear. Thanks for pointing that out.
- Section 3 is highly unclear. There are a lot of wordings which don't say anything, like "a methodology of project and processes", "inductive approach", "solved in sequence" (how else?), "material dimension" (material or dimension?), "functions indicative of the analysis", ..., "finding a low-cost material to receive the information" (which?), etc. Please write down in a clear, concise way what you did, without adding a lot of unnneccessary phrases. This is also valid for the sub-chapters 3.1 ff.
Thank you for this comment. We have reformulated Section 3 on its entirety, changing its subsections and making the text more concise and clear.
- Fig. 2 is unclear. What is, especially, shown in the inset?
Thank you for pointing this out. This figure illustrates the rectangle containing the 3D printed material attached to the cylinder and how it behaved on the adherence tests. The highlighted view shows some of the 3D printed content (lines and shapes) detached from the base surface due to the rotation performed. X corresponds to the distance in a straight line from the detached end of the printed piece to the point where it was still fixed. This explanation was added to the figure caption. The entire test procedure is detailed step by step in the previous paragraph.
- Eq. in line 216:
360° misses a unit.
Thank you for the comment. We have added a small text explaining the angles used in the equation are given in degrees.
A is not a common abbreviation for an angle, why not just alpha?
Thank you for the suggestion. We have replaced A with alpha.
M is not correctly defined - a physical parameter cannot be "measurement", but is apparently a length here.
Thank you for the comment. We have changed "measurement" with "distance", to better represent that it corresponds to the distance in a straight line from the detached end of the printed piece to the point where it was still fixed. We also changed the M letter to X, as shown in the figure.
Phi, on the other hand, is normally an angle, not a diameter (why not just d?).
Thank you for the suggestion. We have replaced phi with d.
Which base material?
Thanks for the comment. We have brought the section that described the substrates used in the experiments to the beginning of Section 3. This way, it will be clear to the reader when we refer to the materials used and their properties.
- After reading the whole section 3, I read a lot of things which are useless (or at least their necessity is not yet clear), such as G-code elements, but on the other hand I have no idea what you really did - I didn't even find the printer, not to talk about the filament or the printed samples, infill density or pattern, and especially the setting of the z-distance which is most important for the adhesion. And what about the substrates you printed on? Normally for a simple 2D printing process, all information about sample preparation fits on max. half a page, and maybe another half page is necessary for the description of the tests.
Thanks for this valuable observation. As said before, the entire Section 3 was restructured. We have removed unnecessary parts such as the G-code elements and added important information such as the 3D printer used.
- line 287: You can definitely not give dimensions like 10.475 mm with 5 significant digits; the printer is not so exact.
Thanks for the observation. We have changed the corresponding value and are now using a single decimal digit for mm measurements.
- "Due to the pressure from the nozzle, it was resized" - by you or in an undesired way? If a second line is added, why are 0.45 mm of the first line + 0.4 mm of the second line = 0.8 mm?
Thanks for the comment. Since we are depositing a single thin line on the 3D printer table, the material spreads and its width grows. When we add a second line, there is a superposition region, so that this extra 0.05 mm is used as "glue" to connect the two or more lines, originating a thicker one. That's why it is possible to have 0.8, 1.2 and 1.6 instead of 0.85, 1.25 and 1.65. We have adjusted the text to make it clearer.
- Fig. 3: The numbers are not readable.
Thanks for the comment. We have increased the image size.
- line 347: Which 2040 measurements?
Thanks for the comment. We have rewritten the respective sentence.
- Fig. 7 finally shows the adhesion test. So why was this roll diameter chosen? And what was the reason not to use any of the common adhesion tests which all the groups working on 3D printing on textile fabrics use?
Thanks for the interesting questions. The new section 3.2 justifies the decision. Basically, we chose to use common materials present in most houses so that the tests could be reproduced without having to acquire specific tools. The used cylinder comes from an aluminum foil cylinder, for instance.
- Fig. 10: Which substrate materials are visible here?
Thank you for such an important question. In figure 10 we have used 75g/m2 sulfite paper. We have added this information to the figure caption and also to other images in order to specify the substrate used as well.
- Fig. 13: Where is the border between both sides of the substrate?
Thank you for the question. Since this specific substrate has different characteristics for its two sides, we have performed two different tests by 3D printing on both sides of the material. According to the figure, one of the sides showed better adherence because of the fibers' interaction to the PLA used.
- line 480 ff: How were the substrate heigths measured?
Thank you for the question. According to the Test Methodology section, "...the average thickness of the surface material on which the experiments would be printed was measured with a caliper at three different points and then added to the height of the first layer to be printed via slicing software (Z Offset option)."
- Fig. 16: Please use English labels. The right axis needs units. It is not possible to connect data if the x-axis is not linearly, logarithmically or similarly spaced (there is simply nothing between EVA and Vergé etc.). What do the red dotted lines mean?
Thanks for pointing that out. We have removed the data connection and also the label on the right side, because it represented the same vertical axis already described on the left side. The red dotted lines are only used to show the pairs of substrates that presented statistically equivalent results.
- Fig. 17-23: Again, please use English labels and, correspondingly, decimal points. And if you want to connect the dots, it is necessary to space the x-axis equidistantly. What to red boxes mean?
Thank you for pointing that out. We have replaced figures 17-23. Red boxes mean that values are statistically equivalent.
- line 619-620: Are the standard deviations small enough to allow for giving angles with 5 significant digits?
Thank you for the question. The angles are calculated based on the formula, and are basically proportional to the measured length and converted using the formula to angles. Because of that, we have median values with two decimal digits.
Fig. 19: Please add the missing units. Why are the four parts not directly labeled with the widths?
Thank you for pointing that out. We have redone Figure 19 and added all units used. Also, we have directly labeled the four parts using the widths.
- Fig. 23: Connections between points are not allowed.
Thanks for pointing that out. We have redone figure 23 and removed the connections between points.
- It is not possible to have a sub-section 6.1 without 6.2.
Thank you for pointing that out. We have removed the title of section 6.1, merging its content with the previous paragraphs of the conclusion section.
- Author contributions are missing.
Thank you for pointing that out. We have added author contributions as requested.
Generally, your work may be interesting, but is is urgently necessary to heavily rewrite the whole manuscript so that the normal scientist has a chance to understand it. I assume you can skip 1/4-1/3 of the text. Use more tables, shorter sentences, write clearly to the point. This is not philosophy, this is materials science.
Thank you for your precious contribution. We have revised all the text. Also, we have changed the structure of some sections and rewritten some paragraphs to improve readers' understanding now. Please let us know if this is enough.

Reviewer 2 Report
see attached pdf

Author Response
Dear reviewer,
Thank you for your careful review of our paper. We appreciate your feedback and comments. We have addressed all of your concerns and have made significant revisions to improve the quality of our paper.
We have addressed the issues you raised concerning paper understanding and organization. We have also re-evaluated our writing using Grammarly. We believe that these revisions have significantly strengthened our paper and have addressed all of the concerns that you raised.
Once again, we appreciate the time and effort you invested in reviewing our paper, and we hope you will find the revised version suitable for publication. Thank you for your consideration. Please let us know if further modifications are necessary. We will try our best to address all of them.
The responses are highlighted in blue. The new PDF of the manuscript highlights the modifications in red.
Sincerely,
João Marcelo Teixeira
The manuscript attempts to present an article about the effectiveness of 3D printing on tactile
content creation for visually impaired users. The article is well written but needs some
additions and structural changes before being able to be considered for publication.
My points are analytically listed below:
Points for consideration:
Point 1: The introduction part about 3D printing is very very small. Add at least one or more
paragraph about its working principles and most important applications. See the following
papers.
Thank you for such important suggestion. An additional paragraph about 3D printing was added to the text in the introduction section.
Point 2: The references number is very low. Consider adding more references in total and
include the following ones in the manuscript.
• 10.5923/j.mechanics.20211001.02
• 10.3844/ajeassp.2022.255.263
• 10.3390/pharmaceutics12020166
• 10.1016/j.promfg.2019.06.089
• 10.1016/j.compositesb.2018.02.012
Thank you for the interesting references. All of them were added to the text.
Point 3: In line 706 and other spots in the text body, better write “3D Printing” than just “Printing”.
Thanks for pointing that out. Most of the occurrences of "Printing" were replaced with "3D Printing" (whenever suitable).

Round 2
Reviewer 1 Report
Dear Authors,
thanks for the improvements. The following minor points have to be changed:
- "beyond the world" - you mean "beyond the earth".
- Generally, please add the citations again which were somehow deleted.
- "verify situations to which the printed images could be subject, as with a sheet of paper in a book" - what is meant here?
- line 235: It must be 360°, else alpha is missing this unit.
The language is understandable.
Author Response
Dear reviewer, once again, thank you for your valuable comments. We are very grateful to them! Please find the responses as follows. The new pdf file has the modified parts highlighted in blue.
Dear Authors,
thanks for the improvements. The following minor points have to be changed:
- "beyond the world" - you mean "beyond the earth".
Thank you for pointing that out. We have fixed the sentence.
- Generally, please add the citations again which were somehow deleted.
Sorry about that, we have upload the complete version now.
- "verify situations to which the printed images could be subject, as with a sheet of paper in a book" - what is meant here?
Thank you for pointing that out. We have replaced the sentence by "The experiments aimed to verify situations similar to real world use. For instance, suppose the tactile content is 3D printed as part of the pages of a children's book. When the pages are manipulated by the children, they are usually deformed and the tactile content must keep adhered to them." to improve understanding.
- line 235: It must be 360°, else alpha is missing this unit.
Thank you for pointing that out. We have added 360° to both sides of the equation.
Reviewer 2 Report
I agree with the revisions, but the authors seem to have forgotten to include the references list from the revised manuscript! Please upload the correct full manuscript.
Author Response
Dear reviewer, once again, thank you for your valuable comments. We are very grateful to them! Please find the response as follows.
I agree with the revisions, but the authors seem to have forgotten to include the references list from the revised manuscript! Please upload the correct full manuscript.
Sorry about that, for some reason the references section was not added. We have upload the complete version now.
Round 3
Reviewer 2 Report
Thank you, I now consent to the article's publication.